# Brain-Derived Neurotrophic Factor Dysregulation as an Essential Pathological Feature in Huntington’s Disease: Mechanisms and Potential Therapeutics

**DOI:** 10.3390/biomedicines11082275

**Published:** 2023-08-16

**Authors:** Andrew Speidell, Noman Bin Abid, Hiroko Yano

**Affiliations:** 1Department of Neurological Surgery, Washington University School of Medicine in St. Louis, St. Louis, MO 63110, USA; speidell@wustl.edu (A.S.); noman@email.wustl.edu (N.B.A.); 2Department of Neurology, Washington University School of Medicine in St. Louis, St. Louis, MO 63110, USA; 3Department of Genetics, Washington University School of Medicine in St. Louis, St. Louis, MO 63110, USA; 4Hope Center for Neurological Disorders, Washington University School of Medicine in St. Louis, St. Louis, MO 63110, USA

**Keywords:** polyglutamine, huntingtin protein, axonal transport, TrkB receptor, neurotrophin receptors, neurodegenerative diseases, epigenetic processes, DNA methylation, histones, targeted molecular therapies

## Abstract

Brain-derived neurotrophic factor (BDNF) is a major neurotrophin whose loss or interruption is well established to have numerous intersections with the pathogenesis of progressive neurological disorders. There is perhaps no greater example of disease pathogenesis resulting from the dysregulation of BDNF signaling than Huntington’s disease (HD)—an inherited neurodegenerative disorder characterized by motor, psychiatric, and cognitive impairments associated with basal ganglia dysfunction and the ultimate death of striatal projection neurons. Investigation of the collection of mechanisms leading to BDNF loss in HD highlights this neurotrophin’s importance to neuronal viability and calls attention to opportunities for therapeutic interventions. Using electronic database searches of existing and forthcoming research, we constructed a literature review with the overarching goal of exploring the diverse set of molecular events that trigger BDNF dysregulation within HD. We highlighted research that investigated these major mechanisms in preclinical models of HD and connected these studies to those evaluating similar endpoints in human HD subjects. We also included a special focus on the growing body of literature detailing key transcriptomic and epigenetic alterations that affect BDNF abundance in HD. Finally, we offer critical evaluation of proposed neurotrophin-directed therapies and assessed clinical trials seeking to correct BDNF expression in HD individuals.

## 1. Introduction

Huntington’s disease (HD) is a universally fatal, autosomal-dominant neurodegenerative disease that affects nearly 40,000 individuals in the United States. This “hereditary chorea,” first fully described and reported by George Huntington in 1872, typically affects individuals in middle age and results in their death within 20 years [1,2]. Despite its debilitating nature in affected individuals, significant breakthroughs in the understanding of HD pathogenesis were not made until the mid-1990s, when the identification of a toxic gain-of-function mutation was first reported [3]. Accordingly, the clinical description of HD has advanced significantly from its initial and primary classification as a movement disorder to a complex neurological condition that affects diverse neuroanatomical substrates and most often includes cognitive and psychiatric symptoms. 

The neuronal atrophy in HD is the consequence of a CAG trinucleotide repeat expansion occurring in exon 1 of the huntingtin (*HTT*) gene on chromosome 4, which encodes a mutant protein (mHTT) with an N-terminal polyglutamine repeat expansion [3]. In HD individuals, an expansion greater than 39 CAG repeats results in 100% penetrance of the HD phenotype, whereas a CAG repeat length between 36 and 39 may result in incomplete penetrance or late-onset of the HD phenotype [4,5]. The N-terminal repeat expansion in mHTT confers a toxic gain-of-function, as this expanded polyglutamine region imparts several new properties to the protein. The most well-known property of mHTT is the ability to form insoluble inclusions with itself and other proteins, but mHTT also disrupts numerous neuron-critical functions in manners independent of aggregate formation. Accordingly, the length of the CAG repeat in *HTT* generally correlates inversely with age-of-onset of HD motor symptoms [6,7,8]. The instability of an expanded trinucleotide repeat, a prominent characteristic seen in HD and other triplet repeat disorders, is associated with disease severity due to expansions in both germline and somatic cells. While the exact mechanisms of repeat instability are not completely understood, various hypotheses have been proposed, involving DNA replication, repair, and gene transcription [9]. Consequently, observations of differential repeat length expansions in the most severely affected neuroanatomical structures may help to explain the regional vulnerability to HD pathogenesis [10,11,12,13]. 

The profound movement dysfunction characteristic of HD is at first the result of the selective loss of subpopulations of medium spiny neurons (MSNs) of the caudate and putamen, two subregions of the basal ganglia that have a shared canonical role in the control of voluntary movements and suppression of involuntary movements. It is near-universally accepted that the initial loss of D2 dopamine receptor-expressing MSNs of the indirect pathway of movement generates the distinctive chorea in the early stages of manifest HD [14,15]. Emerging research also indicates that other factors may affect MSN vulnerability to degeneration in early HD. Recent data-driven approaches indicate that MSN vulnerability may hinge on the position within the striosome–matrix axis [16], the absence or presence of mitochondrial RNA release [17], and the occurrence of somatic repeat expansions [18]. The precise mechanisms underlying increased vulnerability associated with these factors have yet to be elucidated. Whichever factors confer the initial susceptibility to neurodegeneration in HD, the neuron loss that was at first region-specific, eventually progress to broader atrophy of the neostriatum and a majority of telencephalic structures. In fact, the Vonsattel neuropathological staging criteria recognize pathological changes to occur in numerous neuroanatomical regions at later stages of disease progression, including the broader cortical mantle, cerebellum, thalamus, and substantia nigra [19,20,21]. Loss of white matter integrity, believed to be both a cause and result of damage of gray matter neurons, is another pathological feature of HD and can be detected through multiple imaging modalities in HD individuals and small animal models [22,23,24,25]. 

In addition to the movement disorder manifest in HD individuals, it is also well-established that cognitive impairments and affective symptoms are highly co-morbid with HD and often occur prior to the onset of overt HD chorea [26]. This observation is not unexpected, given the role for basal ganglia signaling in higher cognitive function and emotional regulation [27,28]. At later stages of disease progression, more distant neuroanatomical structures exhibit neuronal loss, and the cognitive domains regulated by these neural substrates become altered. For example, volumetric magnetic resonance imaging reports involving HD populations and rodent models consistently detect the loss of gray matter in the hippocampus and other limbic structures [29,30,31]. These atrophic changes are frequently correlated with poorer performance on spatial memory tasks in HD individuals and on hippocampal-dependent behavioral assays in rodent models [32,33,34]. The psychiatric burden in the HD population is substantially higher than for most neurodegenerative disease patient populations, with some observational reports and meta-analyses suggesting lifetime prevalence of psychiatric disorders to exist between 33 and 76% [35,36,37]. These psychiatric disturbances often occur well before the onset of the overt dysfunction of motor control and negatively contribute to quality of life in HD individuals [38]. Exploration of the shared features causal to the motor, cognitive, and psychiatric symptoms that manifest in HD is likely to illuminate common mechanisms amenable to intervention in this patient population.

The examination of molecular features associated with mHTT expression has generated significant insight into specific cytological disruptions present at multiple stages of HD and in corresponding HD models. These include impairments of the regulation of gene expression [39], translation [40], autophagy [41,42], mitochondrial function [43], intraneuronal transport [44], and synaptic function [45]. These impairments possess independent damaging effects on neuronal health and function, but they can also promote detrimental additive effects on a neuron’s ability to withstand the mounting burden of neurotoxic events in HD. As no disease-modifying therapeutic options currently exist for the treatment of HD, the identification and dedicated study of neuron-protective factors are paramount. There is perhaps no greater example of a set of broadly protective factors in the context of neurodegenerative environments than neurotrophins and neurotrophin receptors. In this literature review, we discuss neurotrophins, the compromise of their function in HD, and how their expression or regulation might be manipulated to delay disease progression in HD, with a special focus on the brain-derived neurotrophic factor (BDNF).

## 2. BDNF and TrkB-BDNF Signaling: Critical Intersections with Common Neurodegenerative Processes

BDNF is a member of the neurotrophin family, a small but critically important series of diffusible neuropeptides with countless roles in neuronal development, survival, and function, including synaptic plasticity. Neurotrophins engage major neuronal growth and anti-apoptotic signaling cascades through action at either tropomyosin-receptor kinase (Trk) or Trk-p75^NTR^ heteromeric receptor complexes. For this reason, exposure to neurotrophins, whether constitutive or activity-associated, is strongly supportive of neuron structure and function and can often determine a neuron’s capacity to withstand exposure to neurotoxic stimuli. BDNF and BDNF-responsive neurotrophin receptors are reviewed extensively in this special issue and in recent literature reviews [46,47,48]. Nonetheless, understanding of common features of BDNF signaling outcomes within neurodegenerative processes will serve to illuminate the critical role of BDNF in HD pathogenesis and inform potential therapeutic strategies for the treatment of HD and related conditions.

Landmark studies in multiple neurodegenerative fields have repeatedly demonstrated shared impairments of neurotrophin homeostasis in human subjects and rodent models. These impairments most often include the regional downregulation of BDNF expression in affected neural structures [49,50] or in cerebrospinal fluid (CSF) vs. controls [51,52,53,54]. This downregulation is often associated with the dysregulation of transcription and translation of the *BDNF* gene, but it may also exist as a result of disruption of the proteolysis of its precursor, proBDNF, or through the altered metabolism of its mature isoform [54,55]. To compound the loss of BDNF expression, the trafficking and subsequent release of BDNF is frequently compromised as neural structures degrade, leading to decreased constitutive or activity-dependent release of neurotrophic factors [44,56]. In this way, early loss of neurotrophic support in neurodegenerative diseases, especially BDNF signaling, often feeds into a vicious cycle of molecular events: reduced BDNF signaling impedes the neuronal ability to maintain functional transport systems and synaptic connections, which in turn decreases both the release of BDNF and the response to its presence through its receptors. Nerve growth factor (NGF) and neurotrophins 3/4 (NT3, NT4) also contribute to neuronal survivability, but their action in the mature (i.e., post-development) CNS appears somewhat restricted to special neuroanatomical regions with high expression of their cognate receptors. For example, the loss of supportive NGF signaling in the basal forebrain remains a favored hypothesis for the early and severe regional degeneration of cholinergic cells in Alzheimer’s disease, but NGF alone may be insufficient to explain the degeneration observed in forebrain-innervated regions [57]. Nonetheless, the broader expression of BDNF and greater CNS abundance relative to other neurotrophin family members suggest that the investigation of BDNF dysfunction in neurodegenerative diseases is likely to offer mechanistic insight and fruitful avenues for therapeutic advances. 

The Trk family of neurotrophin receptors has likewise attracted considerable research attention for its role as the major signal transducer system for mature neurotrophins. While the affinity between neurotrophins and neurotrophin receptors is somewhat promiscuous, the strongest bindings (i.e., lowest K_D_) are largely acknowledged to occur between NGF:TrkA, BDNF and NT4:TrkB, and NT3:TrkC complexes. The activation of Trk family receptors is anti-apoptotic in nearly all contexts through the activation of major cell signaling pathways in neurons, including, but not limited to, PI3K/AKT and Ras/MAPK/ERK [58]. These Trk-associated signaling pathways, when activated, also serve to modulate synaptic plasticity [59], neurite regeneration [60], and neurotransmitter release [61]. A fourth member of the neurotrophin receptor family, the p75 neurotrophin receptor (p75^NTR^), forms Trk-p75^NTR^ heteromeric complexes with an exceptionally high affinity for neurotrophins [62]. In an opposite manner, p75^NTR^ also forms a co-receptor complex with sortilin and drives pro-atrophic signaling through the high-affinity binding of proneurotrophins [63]. For this reason, the investigation of changes in Trk abundance is rarely considered in isolation but must be evaluated in light of corresponding changes in p75^NTR^ and pro- and mature neurotrophin expression. The marked and broad downregulation of Trk expression has nevertheless been described in numerous research reports involving major neurodegenerative diseases [64,65,66]. These studies, both in humans and preclinical models, suggest that TrkB loss is often an early event in neurodegenerative disease progression and accompanies the simultaneous downregulation of BDNF [67,68]. Additional reports indicate that TrkB activity may also be inhibited in these progressive disease states via a mechanism involving the direct binding of neurotoxic peptides to internal TrkB epitopes (e.g., alpha-synuclein) or the differential endogenous expression of Trk adaptor proteins [50,69]. Curiously, alternative splicing of TrkB into catalytically inactive “truncated” isoforms (e.g., TrkB.T1) may also act to throttle the broader Trk response to the existing BDNF pool through dominant-negative inhibition. The exploration of the differential expression of full vs. truncated isoforms in models of neurodegenerative diseases represents an exciting research direction, especially in light of reports showing that the manipulation of the TrkB:TrkB.T1 ratio acts to modify disease onset [70,71,72]. 

Finally, the up-regulation of the pan-proneurotrophin receptor p75^NTR^ is a consistent observation in the context of both acute and chronic neural injury [73]. For this reason, impaired processing from the precursor to the mature isoform of BDNF also serves as an explanation for accelerated neuronal death, even when Trk receptor expression remains stable. Mechanisms underpinning the disruption of (pro)neurotrophin processing appear to be more disease-specific than shared, but they may include the downregulation of cellular proteases [55,74] or the modification of proconvertase activity by the expression of inhibiting peptides [75]. Imbalances in the Trk/p75^NTR^ ratio to favor increased p75^NTR^ signaling are also described in multiple research reports, and some of these studies further indicate that this ratio may be affected at a subcellular or synaptic level [64,76]. Whichever mechanisms predominate, the neurotrophin and neurotrophin receptor signaling network exemplifies a dynamic ligand-receptor system, whose disruption has outsized effects on neuronal survival and function in the context of neurodegenerative disease. 

## 3. The Role of BDNF in HD

Since their discovery, neurotrophins have been inextricably linked to neuron viability in the face of both age- and disease-associated pathogenic stimuli. For this reason, impairments of BDNF expression, signal transduction, transport, and metabolism remain fertile targets for investigation within progressive neurological disorders. HD in particular serves as a model neurodegenerative disease in which the normal suites of neurotrophin and neurotrophin receptor functions are compromised. This disruption is repeatedly observed in HD individuals and nearly all available preclinical models; therefore, it is likely to both serve as a critical factor modifying disease progression and simultaneously offer mechanistic insight into major cellular features manifest in HD subjects. 

### 3.1. Loss of BDNF in HD

BDNF expression and biodistribution in Huntington’s brains are dramatically affected in the course of disease progression. Early reports examining BDNF expression in post-mortem tissue from HD individuals found a reduction of up to 82% of BDNF expression in the caudate and putamen vs. controls [77]. Landmark studies by two additional research teams indicated that decreased cortical *BDNF* gene transcription [78] and interruption of the expression of BDNF vesicular transport proteins [79] were both present in HD brains (discussed below). These two findings appear especially critical with respect to HD pathogenesis, as the vast majority of striatal BDNF is synthesized in cortical neurons and subsequently transported to targeted striatal regions [80,81]. TrkB expression is similarly affected in the caudate nucleus and motor cortex of post-mortem tissue from HD brains, with some striking reports indicating up to an 80% loss of TrkB immunoreactivity in HD vs. control brains [82]. In a somewhat contradictory set of findings, the major Val66Met polymorphism in the pro-domain of BDNF, a significant negative modifier of disease progression in several neurodegenerative, affective, and psychiatric disorders, appeared to have no effect on HD pathogenesis [83]. In fact, one report indicated that HD individuals who were heterozygous for this polymorphism exhibited a later age-of-onset than those homozygous for valine or methionine at residue 66, although this effect was only observed in a subset of HD subjects in their study [84]. 

At least two reports indicated that serum BDNF is decreased in HD patients, which appears to correlate inversely with CAG length and the duration of illness, as measured by the unified HD rating scale (UHDRS) [85,86]. At least one ongoing clinical trial evaluated BDNF or TrkB in CSF from HD individuals as a biomarker of disease progression (NCT04012411). More recent reports, however, conflict with earlier observations of decreased peripheral BDNF abundance in HD and further suggest that BDNF alone may not represent as a suitable biomarker for disease progression [87,88]. Ultrasensitive ELISAs do not identify a difference in CSF BDNF between controls and HD individuals, even in advanced-stage HD subjects [88]. This finding is somewhat puzzling in light of the strong positive correlation of measures of disease progression with other CNS-specific biomarkers, including CSF tau [89], mHTT [90], and neurofilament light chain [91,92], but not unexpected given the short BDNF half-life and multiple non-neural pools of neurotrophin expression. The interpretation of plasma or CSF BDNF may be further complicated by the likely compensatory increase in its expression by peripheral cell types. In fact, it was recently reported that a moderate increase in intra-platelet BDNF levels was detected in patients in stage II of HD [93]. Investigation of alternative peripheral pools of BDNF that mirror the decreases occurring in neural structures continues to be of importance in HD [94].

### 3.2. BDNF Interruption in Preclinical Models and Mechanistic Insights

A decline in BDNF expression is not only observed in HD individuals but is reproduced consistently within HD models. In fact, the expression profiling of striatal tissue obtained from two HD models, when compared to humans, indicated that BDNF depletion may represent the most critical role underlying striatal degeneration in HD individuals [95]. In agreement with this finding, rapidly progressive models of HD, such as the R6/2 transgenic mouse, showed an age-dependent reduction in BDNF expression and rescue of TrkB-associated signaling via the administration of recombinant BDNF [96]. The overexpression of BDNF and TrkB in an immortalized cell line model also restored TrkB activity and attenuates caspase-3 activation [97]. Further, striatal BDNF reduction in zQ175 mice was found to exist without corresponding decrements in proBDNF, indicating that the loss of BDNF abundance in HD may involve both transcription-dependent and -independent mechanisms [98]. Interestingly, reduced TrkB activity preceded the loss of BDNF in some of these models, suggesting that impaired TrkB-mediated signaling represents the first event in the cascade of neurotrophin dysregulation events in HD [50,99]. 

Curiously, loss of BDNF is even observed in neuroanatomical structures largely spared by classical HD pathology, although it is unknown if the majority of these observations are restricted to models. In both HD individuals and the N171-82Q model, BDNF expression is sharply reduced in the pontine and medullar regions of the brainstem and was suggested to contribute to accelerated heart rate. Intracerebroventricular administration of BDNF was sufficient to restore the heart rate to normal levels in these HD mice [100]. Disruption of hypothalamic function, believed to generate circadian dysfunction and neuroendocrine abnormalities in HD individuals, may also be related to the suppression of resident BDNF and/or TrkB signaling [101,102]. For example, the breakdown of circadian rest-activity cycles was observed in mouse models and compared well with robust sleep disturbances in HD individuals. The interruption of BDNF-TrkB signaling in the suprachiasmatic nucleus, a hypothalamic subregion responsible for regulating light-dark cycle entrainment, may serve as a contributing factor, especially due to the high local expression of these two proteins [103]. 

Modeling of HD within in vitro systems largely mirrors the decline in BDNF abundance in rodent models and HD individuals. Indeed, mouse primary neuron models that employ the lentiviral transduction of mHTT exon 1 display the downregulation of *Bdnf* gene expression well before decrements in measures of viability [104,105]. Untransfected primary neurons harvested from embryonic-age HD mouse models also displayed aberrations in BDNF expression, transport, or release in culture [106,107,108]. More innovative approaches, including induced pluripotent stem cells (iPSCs) and the direct reprogramming of fibroblasts obtained from HD patients, are increasingly used to model HD [109]. *BDNF* appeared within dysregulated gene networks in some iPSC models [110], and the HD iPSC-derived neural progenitor cells appeared to be more vulnerable to BDNF withdrawal than control iPSC-derived cells [111]. In support of these results, reprogrammed striatal neurons from HD fibroblasts displayed the downregulation of gene expression of the mRNA encoding TrkB vs. striatal neurons reprogrammed from unaffected individuals [112]. Another recent publication employing HD dermal fibroblasts directly reprogrammed to striatal-lineage neurons failed to show differential *BDNF* or *NTRK2* (TrkB) gene expression between HD and control groups, but HD neurons displayed neurite dysmorphogenesis and electrophysiological abnormalities—two neuronal attributes associated with neurotrophin signaling [113]. Finally, limited reports evaluating BDNF endpoints within immortalized striatal neuron models also suggested that neurotrophin abnormalities also exist in these cell lines. STHdh^Q111/111^ striatal cells displayed both reduced expression and subcellular localization of TrkB and impaired BDNF-induced MAPK signaling vs. cells of the STHdh^Q7/7^ control line [114]. Ultimately, the collection of these model systems not only serves to describe the extent of BDNF loss in HD, but it aids in uncovering the set of common mechanisms underpinning BDNF disruption and illuminates the intersections amenable for therapeutic approaches.

HTT’s role as a scaffolder of molecular motor complexes and their cargoes places it in a pivotal intersection to disrupt BDNF’s functions at distal neuronal structures, as anterograde and retrograde axonal transport is critically linked with neurotrophin release and signal transduction. Microfluidic and real-time imaging approaches employing neurons obtained from HD models have been particularly illuminating in examining trafficking deficits in mHTT-expressing neurons. Indeed, a decrease in BDNF travel length and the speed of BDNF-containing vesicles was observed in primary neurons obtained from zQ175 HD model mice [106]. BDNF-specific trafficking impairments were replicated in Hdh^CAG140^ neurons [44] and other models [107]. A correction of transport defects in models is often sufficient to reverse major pathological features [115], indicating that restoring the axonal transport capacity in HD may represent a more attractive and practical approach than the replenishment of the BDNF level more broadly. How might mHTT disrupt axonal transport? HTT is known to serve in a trafficking complex with huntingtin-associated protein 1 (HAP1) and directly facilitates complexing and assembly of dynein and dynactin subunits [116]. Interestingly, strength of the HTT-HAP1 interaction was found to be positively correlated with polyglutamine repeat length, which is likely to impact the trafficking of BDNF- or Trk-laden vesicles [117]. Retrograde transport of TrkB is also negatively affected in a polyglutamine-dependent manner—a finding which has particular implications for the fidelity of TrkB retrograde signal transduction [118]. Finally, the phosphorylation state of mHTT, particularly at serine 421, appeared to modulate the ability of mHTT to bind subunits of motor complexes and transport intracellular cargoes, including BDNF [119]. Phosphoproteomic alterations in neurons, a striking recent finding in pre-symptomatic HD mice, might therefore serve as interesting indirect mechanisms that have knock-on effects on BDNF transport [120]. 

### 3.3. Effects of BDNF Loss on Synaptic Features

There is perhaps no greater purpose associated with BDNF than its repertoire of roles at the synapse, where BDNF signaling is firmly established as a critical factor shaping synaptic development, structure, and function [121]. Loss of synaptic integrity is well-established to underpin deleterious effects on cognitive, behavioral, and motor functions [122]. Consequently, the early observations of aberrant BDNF expression in HD individuals and models quickly led to multiple lines of investigation focusing on synaptic integrity and impairments in synaptic transmission in HD. 

There are likely effects of mHTT on synaptic function that may be mediated by the loss of synaptic BDNF expression or involve additional knock-on effects of altered BDNF-TrkB signaling in HD. Early evidence for a causal role for BDNF in deficits of synaptic efficacy was provided by two groups demonstrating that the overexpression or exogenous administration of BDNF in HD models partially rescues deficits in long-term potentiation (LTP), a major model of synaptic plasticity [123,124]. Xie and colleagues additionally demonstrated that decrements in multiple molecular and behavioral phenotypes in the YAC128 model were ameliorated when crossed with a transgenic line, which overexpressed BDNF in forebrain neurons [125]. Moreover, the pharmacological induction of BDNF through α-amino-3-hydroxy-5-methyl-4-isoxazolepropionic acid receptor (AMPAR) modulators was sufficient to rescue LTP deficits and activity-driven actin polymerization at the dendritic spine in a CAG140 mouse model [126]. The restoration of plasticity-associated endpoints via the exogenous application of neurotrophins is a Trk isoform-dependent process and appears to be a unique property of BDNF among neurotrophins [127]. The stimulation of adenosine-2A receptors, believed to directly activate TrkB receptors and potentiate the BDNF-mediated modulation of synaptic transmission, also appeared to restrain excitotoxicity in R6/2 rodents [128,129]. The expression of mHTT is not merely associated with reduced plastic capacity within LTP paradigms, but also affects the activity-dependent release of BDNF into the synaptic cleft, at least at corticostriatal synapses. A unique approach involving the real-time sensing of BDNF release revealed the interruption of fusion modes of BDNF-laden vesicles following the high-frequency stimulation of Hdh^Q175^-expressing cortical projection neurons [130]. Finally, mHTT expression may also be associated with the maladaptive recovery of synaptic function after neural injury. A recent study by Spies et al. indicated that induced dsDNA breaks in the STHdh^Q111/111^ cell model had differential effects on gene expression associated with synaptic plasticity vs. controls, including a lack of a compensatory *Bdnf* and *Arc* upregulation [131]. 

There is limited but exciting evidence that the synaptic expression of neurotrophin receptors is similarly compromised in HD. Reduced expression of TrkB in HD mouse models and in HD individuals was first reported nearly two decades ago, although it was not clear whether this decrement was neuron-wide or specific to synaptic terminals, a critical site of TrkB expression [82]. Evidence of disrupted TrkB retrograde vesicular transport in striatal dendrites was shown to exist alongside reduced ERK and c-Fos activation through a unique microfluidic approach [118]. Nearly simultaneously, Brito and colleagues reported the upregulation of p75^NTR^ at the synapse of hippocampal neurons in two mouse models of HD [132]. This upregulation of p75^NTR^ therefore likely co-exists with decrements in synaptic TrkB expression and favors the activation of pro-atrophic/pro-apoptotic neuronal signaling cascades in humans and models [76]. When considered alongside BDNF-related abnormalities, expression and localization changes in these BDNF-responsive receptors likely exacerbate deficiencies in neurotrophin signaling associated with mHTT expression.

It is important to note that various research groups have identified roles for HTT at the synapse that are independent of BDNF, including the scaffolding of multiple cytoskeletal and synaptic vesicle proteins and the regulation of vesicle endocytosis [133,134,135]. More recent reports point to emerging roles for HTT in the modulation of vesicular protein–protein interactions, presynaptic homeostasis, and activity of terminal secretases [133,136,137]. Unsurprisingly, HTT was even shown to be required for short and long-term potentiation at Schaffer collaterals, emphasizing the importance of synaptic HTT expression [138]. Therefore, the loss of a functional *HTT* allele in HD likely contributes to the simultaneous exacerbation of decrements of BDNF expression and neurotrophin signaling at the synapse. In agreement with this observation, many pathological processes are reversed by the overexpression of wild-type HTT protein, indicating that loss of normal HTT function in adulthood may promote an equally damaging outcome as the expression of its mutant counterpart [139,140].

## 4. Mechanisms of *BDNF* Gene Regulation in HD

### 4.1. HTT-Associated Regulation of Bdnf Transcription

A central molecular feature of HD pathogenesis is the robust transcriptional dysregulation observed in affected cells, which occurs even in premanifest stages of HD [39,141]. For this reason, mechanisms favoring the altered regulation of gene expression have provided multiple ripe targets for HD investigators over the last two decades. As a direct result of the -omics era explosion of analytical techniques, details of *HTT*/*mHTT* gene regulatory networks have now grown to staggering scales. Wild-type HTT is known to interact with several transcription factors (TFs) [142], but a mechanistic explanation for transcriptomic dysregulation involving HTT loss-of-function as a scaffolder of transcription complexes is not yet supported. In fact, a study employing neural tissue from various HD models did not show a main effect of wild-type *HTT* gene dosage on the degree or number of transcriptional alterations [143]. Instead, mHTT likely adopts a toxic gain-of-function through the sequestration or segregation of TFs within inclusion bodies. In support of this hypothesis, there exist several reports showing that the induced expression of protein fragments with polyglutamine expansions is sufficient to sequester major gene regulatory proteins within nuclear inclusion bodies [144,145]. Alternatively, given that HTT and mHTT occupy different genomic loci, existing gene regulation machinery may be routed to disparate genomic regulatory regions by mHTT and inappropriately suppress or permit the synthesis of corresponding transcripts [146,147]. The decreased transcription of *BDNF* is a central feature of HD molecular pathology [78]. Consequently, gene regulation of this neurotrophin provides an ideal model for the examination of genes whose expression is interrupted in HD. 

The genomic structure of the *BDNF* gene in both rodents and humans is extraordinarily complex [148]. This intricate genomic architecture, however, permits the expression and distribution of *Bdnf* transcripts by neurons in precise spatiotemporal patterns [149]. At least nine exons exist within the *Bdnf* gene. The *Bdnf* architecture includes multiple 5′ noncoding exons and one exon containing the pre-proBDNF coding sequence. Each of these 5′ non-coding exons is linked to separate promoters. Additionally, multiple untranslated regions downstream (3′-) of the *Bdnf* coding sequence are known to be involved in the regulation of transport of the mature transcript [150,151]. This complex exon-promoter scheme allows for a remarkable degree of control of expression of gene regulation, but it also serves as a critical vulnerability in disease states involving the altered regulation of gene expression; at least three exons (exons II, IV, and VI) are known to be downregulated in human HD [78,152,153]. *BDNF* gene expression in neurons is partially, but directly, regulated by wild-type HTT through its interaction with a transcriptional repressor, the repressor-element 1-silencing transcription/neuron-restrictive silencer factor (REST/NRSF). Wild-type HTT sequesters REST/NRSF in the cytoplasm, preventing it from binding to a corresponding response element or neuron-restrictive silencer element (RE1/NRSE) within promoter II of *Bdnf*, thereby facilitating *Bdnf* transcription [152,154]. Mutant HTT, on the other hand, fails to sequester NRSF/REST, accelerating its translocation to the nucleus where it binds to the *BDNF* promoter and represses its transcription (Figure 1A) [155]. In support of this mechanism, the expression of a REST decoy oligonucleotide in a cellular model of HD was sufficient to alter the REST occupancy of its response element in the *Bdnf* gene and rescue *Bdnf* expression [156]. Notably, mHTT interacts with the cAMP response element-binding protein (CREB) binding protein (CBP). CBP is a transcriptional coactivator with intrinsic histone acetyltransferase (HAT) activity that binds to CREB, a transcription factor whose phosphorylation is a direct downstream outcome of TrkB activation [157]. CBP was demonstrated to be sequestered to mHTT intranuclear inclusions in HD mice [158,159]. In this way, mHTT could serve to both suppress *Bdnf* gene expression and diminish the neuronal response to existing BDNF-TrkB signaling through CBP/CREB. Finally, it is important to recognize that mHTT is able to bind directly to DNA and occupy genomic regulatory regions in vivo in a polyglutamine-dependent manner [146,147], providing a non-specific but key mechanism by which *BDNF* transcription could be negatively affected.

In addition to mHTT-associated alterations in the rate of *Bdnf* transcription, altered regulation of the transport of the *Bdnf* transcript itself may be compromised in HD. Discrepancies in mRNA transport are likely to have outsized effects in neurons, especially considering that the local and rapid translation of the *Bdnf* transcript is of high importance for the structure and function of dendritic spines and other distant structures. Intraneuronal *Bdnf* mRNA transport is orchestrated through regulatory regions within at least two distinct promoters and the 3′UTR regions of its gene body and could therefore be affected by broader patterns of regulatory disruption discussed above [151,161]. *Bdnf* mRNA targeting to either somatic or dendritic compartments is determined through polyadenylation states within its 3′ regulatory region [151]. Consequently, aberrant polyadenylation may serve as a contributing factor for the disruption of *BDNF* transcript targeting in HD, especially given recent reports of shared polyadenylation abnormalities across major neurodegenerative disorders [162]. Although no significant evidence exists to support the specific disruption of trafficking of the *Bdnf* transcript in HD, we note that others have shown that BDNF and its transcripts are co-segregated in neurons [161]. Therefore, reports of altered BDNF protein distribution and transportation across degrading trafficking networks within HD neurons may be equally applicable to its transcript [79,108,118].

### 4.2. Epigenetic Disruption of BDNF Expression

There exist many means by which epigenetic mechanisms can affect gene expression. Laying “above” (epi-) the genetic sequence encoded within the DNA itself, this series of transcriptional control mechanisms serves to affect gene transcription in both heritable and non-heritable manners. Histone post-translational modification (PTM) and DNA methylation, two forms of epigenetic regulation, have both been connected to altered expression of genes in HD. Therefore, the robust and profound disruption of gene expression in HD may arise from the failure of epigenetic control in affected neuronal populations. A greater understanding of epigenetic regulation and its failure in HD illuminates several potential areas for research and therapy.

Histone PTMs are the set of covalent alterations that occur at certain residues on the N-terminal tails of histone octamers, the core structural components of nucleosomes. These reversible modifications, which may include acetylation, ubiquitination, methylation, phosphorylation, and SUMOylation, serve to define gene expression [163]. For example, histone PTMs represent readable markers for epigenetic effector complexes and subsequently alter local chromatin states to define gene expression. Trimethylation (me3) at lysine (K) 27 of histone H3 (H3K27me3), a well-known repressive histone mark, serves to compact nearby chromatin through action of the polycomb-repressive complexes [164]. Alternatively, the ionic properties of these PTMs are also believed to broadly affect the affinity of the histone-DNA interaction and may affect gene transcription via steric accessibility and electrostatic shielding effects. In general, the repressing or activating properties of histone marks generally depend on the amino acid residue, which is modified, the type of modification, and their genomic locations, including their proximity to transcriptional start sites. 

How might histone modifications be altered in HD to contribute to gene expression dysregulation, including that of *BDNF*? Research examining histone PTM ratios in humans and models has consistently reported the differential expression of both repressive [165,166] and activating [167,168,169,170] histone marks. The altered expression or activity of the enzymes that catalyze these histone PTMs may thereby affect PTM abundance or destabilize the ratio of repressive vs. activating marks. Inhibition of the activity of histone-modifying enzymes or the removal of the modification(s) that they deposit, however, does not consistently appear to rescue the loss of *BDNF* transcription in HD, even when the expression of other neuron-critical genes is restored. A study has reported the enrichment of an active histone PTM (H3K4me3) at *BDNF* regulatory regions in mouse and human brains, which were downregulated in the HD context, and a corresponding rescue in gene expression was observed when the demethylase catalyzing this mark was inhibited in a Drosophila model [169]. The acetylation of H3K27 is associated with *Bdnf* promoter occupancy and regulates *Bdnf* transcription [171]; however, only limited studies involving this and other marks have been conducted in HD models and humans, and these studies did not specifically evaluate *BDNF* expression [167,168]. The inhibition of histone deacetylases (HDACs) might also have indirect benefits on *Bdnf* expression by rescuing transport deficits through non-epigenetic mechanisms that target non-histone substrates. For example, HDAC6 inhibition, by increasing the levels of α-tubulin acetylation, has been demonstrated to rescue axonal transport deficits in other neurodegenerative diseases and was recently evaluated in two HD preclinical models [172,173]. The lack of additional direct evidence in this direction may indicate that the presence or absence of a particular histone PTM has only modest effects on *BDNF* expression in HD models. Indeed, both focused molecular and larger-omics approaches evaluating transcriptomic endpoints under the non-selective inhibition of HDACs failed to detect (or did not report) changes for *Bdnf* expression [174,175,176]. More likely, changes in histone PTM distribution and abundance alone do not affect *Bdnf* transcription, but act via the engagement of other gene expression mechanisms, including via coupling with alternative epigenetic regulatory processes. In fact, gene expression analyses of progeny from HDAC inhibitor-treated male HD mice showed improved HD phenotypes vs. vehicle controls through the mechanism involving altered DNA methylation patterning [177]. The inhibition of the families of enzymes that catalyze histone PTMs may ultimately prove to harbor beneficial effects via crosstalk between epigenetic regulatory modes. In this way, histone PTMs and DNA methylation cooperate to regulate gene expression to establish stable or dynamic chromatin states.

Methylation of the fifth position of the cytosine ring (5-methylcytosine or 5mC) is a well-characterized epigenetic mark that is associated with gene repression when present at promoters and regulatory regions. Approximately 75% of the CpGs (cytosine-guanine dinucleotides) in neurons possess the 5mC mark [178]. “CpG islands”, the unmethylated genomic regions present around regulatory regions of active genes, typically do not exhibit this mark. Thus, methylation states of promoters affect the binding of regulatory complexes to define the rate of gene transcription. As a consequence, aberrant methylation patterning can affect gene expression in pathological states. DNA methyltransferases (DNMTs), the set of enzymes that catalyze the addition of this methyl group, are present in neurons and can be active in maintenance or de novo methylation. Therefore, changes in the global methylation landscape and the up- or down-regulation of methyltransferases may both serve as mechanistic descriptions for transcriptomic aberrations in HD. In support of this hypothesis, early studies in this direction employing reduced representation bisulfite sequencing and methylated DNA immunoprecipitation (MeDIP)-seq indicated differential methylation at promoter regions in striatal cells from an HD model vs. wild-type controls [179]. Profound cytosine methylation changes also appear to be associated with mHTT expression in rodent and large animal models and HD individuals [180]. The complex genomic structure of *Bdnf* includes multiple promoters, and at least promoter IV is differentially methylated in HD model cortical neurons vs. controls, providing a putative but direct explanation for dysregulated *Bdnf* transcription (Figure 1B) [104,105]. In this model, the critical activity-associated promoter IV is directly methylated by neuronal DNMTs, while promoter VI remains unmethylated, indicating that promoter VI is not a direct target of DNA methylation [104]. Indeed, the pharmacological inhibition of DNMTs restored methylation at promoter IV of *Bdnf* and its transcription in vitro as well as rescued transcription of critical HD-associated genes in R6/2 mice [104]. This effect was also reproduced with the RNAi-mediated knockdown of DNMTs in HD models, suggesting a direct role for DNMTs in the hypermethylation of downregulated genes in HD. Altered *BDNF* promoter IV methylation in HD is not restricted to neurons in affected neuroanatomical structures, but it is also observed in non-neural cell types and may contribute to alter peripheral pools of BDNF to exacerbate the central loss of this neurotrophin [94,181]. Several intriguing reports assessing epigenetic endpoints in serum or post-mortem samples from humans suffering from other neurological and psychiatric disorders also observed the differential methylation of *BDNF* promoters [182,183,184]. The precise mechanisms by which DNMTs are inappropriately routed to specific promoters in HD are still unclear and require further investigation. It is important to consider that readings of the set of cytosine modifications in HD may also exist as contributing factors in epigenetic disruption. Intriguingly, HTT was found to possess a direct binding property to Methyl-CpG Binding Protein 2 (MeCP2), a critical scaffolding protein that serves as a reader of 5mC [160]. This property was shown to strengthen in a polyglutamine repeat-dependent manner, which altered the binding of MeCP2 to promoter IV of *Bdnf*, and resulted in its downregulation in the STHdh^111/111^ cell model of HD (Figure 1B). 

5mC is not removed directly in post-mitotic cells but via a series of ten-eleven translocation (TET1)-dependent enzymatic conversions which include 5-hydroxymethylcytosine (5hmC), 5-formylcytosine, and 5-carboxylcytosine [185]. These alternative methylation states also appear to have gene-repressive properties at promoter regions; however, they are recognized by different reader proteins, and their roles in pathological states remain unclear. Landmark studies nearly 15 years ago established that 5hmC expression is highest in the central nervous system and comprises between 0.3–0.7% of all nucleotides in neural tissue [186,187]. Intriguingly, research teams assessing genome-wide 5hmC maps in other pathological states have reported alterations in 5hmC abundance within critical gene bodies, including that of *Bdnf* [188,189]. A study evaluating post-mortem HD striatal tissue observed altered 5hmC abundance in the adenosine A_2A_ receptor (A_2A_R) gene body. This outcome was recapitulated in the R6/1 and R6/2 models, which correlated with decreased striatal expression of A_2A_R in these mice [190]. This finding is particularly intriguing, given that A_2A_R is a receptor whose activity is known to maintain BDNF levels in the CNS and can stimulate TrkB expression [191]. In fact, it appears that the genome-wide loss of 5hmC in the mouse brain was a key feature of at least one HD model (YAC128) at multiple ages [192]. Prenatal 5hmC loss may impart legacy effects in HD as a result of 5hmC’s proposed role in the neurodevelopmental control of gene expression [193]. In support of this observation, Barnat et al. recently showed that mHTT alters human neurodevelopment at even embryonic stages [194]. The extent to which 5hmC abundance is altered within the *BDNF* gene throughout developmental timeframes remains an unexplored research question that may have exciting implications, given the wide array of BDNF neurodevelopmental functions.

### 4.3. Other Putative Regulatory Modes for BDNF Expression

Still, other intriguing explanations have been proposed for the decrements in *BDNF* gene expression in HD. Splicing of pre-mRNAs into mature transcripts has only recently been shown to be broadly dysregulated in HD, an observation which could have a significant impact on *Bdnf* and genes with similarly complex genomic structure. Research reports detailing exon skipping, cryptic exon exposure, and intron retention are not infrequently reported in HD brains and models [195,196,197]. Intriguingly, the *mHTT* gene itself (but not wild-type *HTT*) is mis-spliced in HD individuals and knock-in models, which results in the synthesis of a transcript encoding only the pathogenic exon 1 fragment [198,199]. This finding aligns with the significant disruption of splicing factor expression observed in HD and the observations of profound splicing errors in other neurodegenerative diseases. To date, no splicing aberrations have been reported with respect to the *Bdnf* transcript in HD models. A growing body of research, however, has illuminated mHTT inclusion sequestration and mis-localization of the activity of the well-known splicing factor transactive response DNA-binding protein of 43kDa (TDP-43) in HD [200,201]. TDP43 was recently linked with impaired activity-dependent BDNF secretion, and it’s interruption in HD may offer a causal explanation for altered BDNF regulation through aberrant splicing [202].

MicroRNAs (miRNAs), owing to their critical post-transcriptional regulatory role, have come under particular scrutiny for a putative role in *BDNF* silencing [203,204]. Some reports indicate that up to 85 distinct deregulated miRNAs may exist in human HD brains [205,206]. A unique hybrid computational approach by Muller further indicated that *BDNF* lies at a critical hub of the miRNA-mRNA regulatory network formed by these miRNAs and, intriguingly, that two of these miRNAs (miR-10b-5p and miR-30a-5p) have putative targets in the 3′UTR of the *BDNF* transcript [207]. The upregulation of miR-10b-5p and miR-30a-5p, also observed in HD brains, was further predicted to strongly repress *BDNF* expression through an in silico approach but has not yet been demonstrated in living systems [207]. Other miRNAs may have indirect, but beneficial, effects on *Bdnf* expression in HD models through the restoration of other modes of gene regulation discussed above [203]. Antisense inhibition of *BDNF* transcription can also serve to restrain *BDNF* transcription in vitro and in vivo through the expression of long non-coding natural antisense transcripts (NATs). Knockout of the NAT for *Bdnf* can enhance cognitive function through the disinhibition of the antisense-mediated decay of the *Bdnf* transcript, but the manipulation of NAT expression has not yet been attempted in the context of HD [208,209]. 

## 5. BDNF-Focused Therapeutic Approaches for HD

To date, no single therapeutic strategy has shown to be effective in the reduction or delay of classical signs of HD progression in humans. The existing tetrabenazine-based FDA-approved pharmacotherapies for use in HD largely target the reduction of choreic movements in HD through the inhibition of the vesicular monoamine transporter 2 (VMAT2), but they are insufficient to mitigate HD pathophysiology [210]. The identification of additional druggable targets, including those directly causal to disease pathogenesis, is therefore desperately needed for the treatment of this devastating disease. 

### 5.1. BDNF Supplementation

For some time, the utilization of BDNF supplementation or direct treatment has been considered as a therapeutic approach for addressing neurodegenerative disease. Accordingly, many of these approaches have been evaluated in HD models. Limited preclinical research indicates that chronic systemic BDNF delivery through indwelling osmotic pumps increased the synthesis of BDNF in the CNS and diminished some measures of disease progression in R6/2 mice [96,211]. A study by da Fonseca and colleagues demonstrated that intranasal administration of BDNF alleviated anhedonic and depressive-like behaviors at early disease stages in the YAC128 model and further stimulated hippocampal neurogenesis [212]. Unfortunately, the systemic administration of BDNF is unlikely to represent a viable neurotrophin-focused strategy in humans, as factors related to blood–brain barrier permeability, short half-life, and off-target effects in the peripheral nervous system and other structures preclude its effectiveness in vivo [213,214]. Other research teams have instead employed viral vector approaches for the long-term CNS-specific expression of BDNF or other neurotrophins. An adeno-associated viral (AAV) vector encoding *Bdnf* cDNA under the control of an astrocytic promoter, when injected into the striatum of R6/2 mice, was shown to delay the onset of HD-specific behavioral abnormalities. This effect may be related to increased neostriatal neurogenesis rather than the reduction of MSN death, as BDNF overexpression through AAV approaches has been shown to increase doublecortin immunoreactivity in R6/2 mice [215]. Other neurotrophins, when overexpressed, may support or potentiate the activity of BDNF. In the quinolinic acid rodent model of HD, rats that underwent the unilateral intrastriatal injection of a BDNF or Glial-derived neurotrophic factor (GDNF) AAV vector were protected against the loss of critical interneuron subpopulations affected early in this model [216]. It is important to note that, aside from translational issues concerning invasiveness and the fine control of gene expression in vivo, direct BDNF synthesis via viral vector approaches may also exert harmful neural effects in humans that are not observed in models with short lifespans. For example, it is unlikely that impairments of BDNF transport or metabolism will be directly rescued by BDNF administration alone. Increased and sustained activation of p75^NTR^ in the context of reduced TrkB expression may instead exacerbate disease-specific outcomes in HD patients and disease models. For these reasons, these “direct” strategies have largely failed to gain practical success outside of preclinical models.

Engineered mesenchymal stem cell (MSC) transplant is under growing consideration as a therapeutic agent in HD for their neural repair properties, which are partially connected with their constitutive ability to release BDNF. At least three groups have reported that the intravenous treatment of MSC-like human dental pulp stem cells possesses the ability to alleviate key HD-like pathological features in the 3-nitropropionic (3-NP) rat HD model [217,218,219]. One of these groups illustrated increased BDNF immunoreactivity in the striatum of these 3-NP/MSC treated rats vs. controls [219]. Other groups described the mitigation of striatal atrophy and an extended lifespan in two additional HD models (YAC128 and R6/2) that were engrafted with intrastriatal human MSCs, although this study required the use of mice on immunocompromised backgrounds [220]. Nonetheless, Cellavita, a MSC-based therapy involving the regular systemic administration of human MSCs, is currently in a phase II clinical trial in Brazil for HD (NCT03252535). More sophisticated transplant and genetic expression strategies may prove to be more beneficial. The direct transplant of human neural stem cells into the striata of R6/2 and Hdh^Q140^ mice resulted in improved motor and electrophysiological deficits in these models, thought to be facilitated by increased BDNF production [221]. One report, involving the engraftment of conditionally immortalized neural stem cell lines CTX0E03, documented increased counts of DARPP32- and BDNF-positive cells in the striatum of the quinolinic acid (QA) rat HD model. These cells appeared to have beneficial effects on inflammation markers on neighboring neurons, suspected to be related to BDNF [222]. Another report found that the intracerebral transplantation of BDNF-overexpressing human neural stem cells into QA-lesioned rats reduced the striatal expression of molecular markers of inflammation [223]. It is worth noting that the use of MSCs and other stem cells in transplantation-based approaches comes with its own sets of limitations. These include delivery concerns, tumorigenic potential, viability in vivo over extended timeframes, immunogenicity, and heterogeneity of these cells [224,225]. 

More indirect and less-invasive experimental manipulations that serve to broadly up-regulate BDNF may alleviate disease progression and offer immediate insight for the clinical management of HD in humans. Exercise enrichment, a modification which is well-connected to a beneficial CNS-wide BDNF response, was sufficient to rescue the loss of BDNF in the hippocampus and striatum of R6/1 mice [226]. Accordingly, the recent ENGAGE-HD (ISRCTN65378754) and PACE-HD (NCT03344601) clinical trials employed physical activity interventions in HD individuals but did not evaluate BDNF expression as primary or secondary measures of interest. Unfortunately, while non-control participants reported greater levels of physical activity and fitness, these studies reported no effect for activity on the UHDRS motor score. It is important to note that physical activity interventions may only be possible in pre-manifest and early manifest stages due to the profound loss of movement control in HD, thereby limiting their therapeutic potential. Environmental modification and enrichment (EME) also appear to bolster BDNF expression in preclinical models. Frequent introduction of novel objects into the home environments of R6/1 strongly upregulated cortical BDNF expression and delayed the onset of motor symptoms in this model [226]. Intriguingly, physical-activity and EME appeared to differentially modify BDNF localization in an exon-specific manner in R6/1 mice, pointing to discrete mechanisms that have a shared outcome of enhanced BDNF signaling [227]. Other EME interventions in HD individuals such as music, language, and multidisciplinary/hybrid tasks have suggested modest benefits for these forms of cognitive and social stimulation [228,229,230,231]. As above, results of human HD trials involving these and other environmental modifiers are mixed, and it is still unclear if the effects of these limited interventions are related to increased BDNF expression in the CNS. Ultimately, future management of HD will likely involve combinations of physical activity and other interventions alongside emerging pharmacotherapeutics.

### 5.2. BDNF-Targeting Therapeutics

The repurposing of emerging small molecule compounds, especially those aimed at alleviating impairment in key cellular pathways, may also indirectly up-regulate BDNF or its signaling and therefore offer similar promise to more direct pharmacotherapies in HD. For example, at least two groups have evaluated the metabotropic glutamate receptor 2/3 (mGluR2/3) agonist LY379268 in preclinical HD models for its enigmatic BDNF-stimulating property [232,233]. Some of these reports confirmed the up-regulation of *Bdnf* transcription [232]. Other groups attempted to employ small molecules to indirectly supplement a BDNF-TrkB signaling tone otherwise lost in disease progression. The A_2A_R agonist inosine mitigated 3-NP-induced motor abnormalities and body weight loss by increasing BDNF signaling through the TrkB-ERK-CREB pathway [234]. Another study showed that the activation of sphingosine-1-phosphate receptor 5 (S1PR5) by the agonist A-971432 delayed the onset of motor dysfunction in R6/2 mice by the activation of Trk-downstream AKT and ERK pathways [235]. Other small molecules, including those with action at TrkB [236,237], p75^NTR^ [238], and REST [239] have also shown limited beneficial effects in HD models, likely due in part to the partial rescue of BDNF expression or TrkB activity through their pharmacological targets. 

The use of the sigma-1 receptor (S1R) agonist pridopidine is under increasing consideration for the treatment of HD. Pridopidine was shown to promote a significant increase in BDNF axonal trafficking, TrkB signaling at the synapse, and enhanced release of glutamate, albeit within a reconstituted “brain-on-a-chip” model of HD [44]. A bulk RNA-seq approach in the YAC128 model also indicated the rescue of neurotrophin expression associated with pridopidine treatment [240]. It is still unknown by which mechanism(s) the activation of the sigma-1 receptor results in the restored trafficking of BDNF, but it is suggested that the reduction of ER stress and/or stimulation of proteolytic degradation of mHTT aggregates may have beneficial knock-on effects on axonal transport or expression in this model [241,242]. Nevertheless, the use of pridopidine in HD has advanced into HD-focused clinical trials. Of note, the PRIDE-HD phase II exploratory trial (NCT02006472) documented a delay in the reduction in total functional capacity in HD patients over one year of pridopidine treatment, but is not known if BDNF up-regulation was responsible for this outcome [243,244]. Pridopidine is currently in a multicenter phase III clinical trial for patients with early-stage HD; however, this trial will not evaluate BDNF expression nor non-neuropsychological outcomes as part of its aims (NCT04556656). Another indirect pharmacological strategy, inhibition of phosphodiesterase 4 (PDE4), was also proposed as a therapeutic approach for its ability to stimulate neural BDNF expression. To date, two phase I clinical trials have taken place that evaluated safety and tolerability of the PDE4 small molecule inhibitor GSK356278, with the eventual goal of use of this compound in HD (NCT01602900, NCT01573819). It is not known whether GSK356278 provoked an increase of CSF BDNF levels in these studies, and it has yet to advance to phase II trials for use in HD.

Still, other research terms have engineered novel peptidergic compounds that specifically modify BDNF expression. Two intriguing reports from Couly et al. indicated that a 23 amino acid peptide “P42” constructed from HTT, when delivered systemically over several weeks, was sufficient to rescue deficits in Schaffer collateral LTP and had modest effects in restoring motor function in the R6/2 model. These effects were suggested to be related to the up-regulation of BDNF and increasing activity at TrkB in P42-treated mice, although it is still unclear the set of cellular mechanisms by which this effect was produced [245]. Somewhat encouragingly, the positive effects of P42 are potentiated by habitat enrichment in this model, an environmental modification that is well-connected with a beneficial BDNF response [246]. Another group reported that the intranasal administration of a recombinant vasoactive intestinal peptide (VIP) upregulated striatal BDNF in the R6/1 model, likely through coupling with the cAMP–PKA–CREB pathway (Figure 2) [247]. In summary, advancements in neurotrophin drug discovery and delivery will unlock previously inaccessible therapeutic options for the treatment of HD and various neurodegenerative diseases.

## 6. Discussion

The dysregulation of BDNF is a well-documented and significant molecular alteration in HD and plays a critical role in its pathogenesis. Substantial evidence suggests that reduced BDNF trophic support and impaired TrkB signaling underlie the degeneration of striatal projection neurons, the most vulnerable cell type in HD. A deeper understanding of the mechanisms leading to deficits in BDNF-TrkB signaling is essential for developing effective neuroprotective treatments for HD. 

Despite continued research into neuron-protective factors for use in neurodegenerative diseases, neurotrophin-based therapeutic approaches have yet to gain serious attention in clinical practice. The set of challenges restraining the identification and development of CNS pharmacotherapeutics is exemplified by BDNF. These practical hurdles include poor half-live in vivo, off-target effects, and poor blood–brain barrier (BBB) penetrability. Drugs that aim to target BDNF expression in vivo must also overcome these obstacles. Still, there currently appears to be more promise for neurotrophin-based therapies in HD than in recent history, owing to advances in the understanding of BBB physiology. Improvements in CNS drug delivery methods, including transcranial focused ultrasound-induced permeabilization [248,249], nasal atomization and insufflation [250,251], and novel drug carriers such as immunoliposomes [252] and nanoparticles (NCT03815916, NCT03020017), have all recently shown positive results in humans for CNS disorders or in animal studies. Continued advances in CNS drug delivery will therefore always be of critical importance in HD, the neurotrophin field, and neurodegenerative research more broadly.

Due to the complexities involved with neurotrophin drug delivery, and the monogenic nature of HD, many compounds investigated within the HD clinical trial pipeline to date did not target BDNF or TrkB directly, but instead acted on mHTT expression itself. It is suspected, but not yet established, that the downregulation of mHTT or reduction of mHTT aggregation through RNA interference may indirectly increase *BDNF* expression in neurons and thereby provide benefits to neuronal health [130]. The administration of antisense oligonucleotides (ASOs) against HTT or mHTT transcripts therefore remains a popular therapeutic approach. Recent and ongoing ASO clinical trials undertaken by Ionis Pharmaceuticals (NCT02519036, NCT03761849, NCT05686551) and Wave Life Sciences (NCT03225846, NCT03225833, NCT05032196) unfortunately have shown modest or no improvement in measures of disease progression. Similar discouraging results from repurposed oral HTT-splicing modulators, such as Branaplam, were recently reported (NCT05111249). It is unclear if CSF or serum BDNF were evaluated within these trials, making comparisons with preclinical research in this direction difficult. It is also important to note that some studies have suggested that the presence of mHTT inclusions is instead a neuronal resiliency mechanism which serves to impede other toxic gain-of-functions of mHTT [253]. HTT-lowering strategies may benefit from the more precise and mutant allele-specific inhibition of *HTT* targeting, rather than the broad silencing of wild-type and mutant *HTT* transcripts, given the growing repertoire of observed wild-type HTT neural functions related to BDNF. The continued improvement in allele-selective ASO design and delivery will undoubtedly have considerable implications for HD.

In the present review, we illustrated numerous ways in which BDNF expression, transport, and release are compromised in HD. However, there remains much to be uncovered in relation to the intersection of BDNF dysregulation and HD. For example, questions revolving around the ability of BDNF to modulate CNS inflammation associated with HD remain underinvestigated. The nature of the glial inflammatory response in the presence of reduced BDNF in HD may be an especially important research direction because some neurotrophins, including BDNF, appear to temper the glial inflammatory response to favor anti-inflammatory modes of activation in certain models [254,255,256]. The broad use of BDNF-directed therapies may also have undesired effects on other cell types and within different neuroanatomical substrates, which provides an incentive for the selective regional targeting of BDNF-based treatments. The development of therapeutics and future clinical management of HD will therefore benefit from considering the effect of BDNF loss on diverse neural and non-neural cellular functions. Ultimately, despite stubborn challenges to their use in the CNS, BDNF-directed pharmacotherapies may represent a desperately needed disease-modifying treatment for HD.

## 7. Conclusions

HD remains an incredibly challenging neurodegenerative disease to address because of the paucity of available molecular targets and lack of suitable therapeutic intervention strategies. Although a few FDA-approved pharmacotherapies exist to manage the classic choreic symptoms coupled with HD, none are able to slow or prevent HD pathogenesis. Therefore, candidate approaches involving broadly protective factors may prove to be crucial disease-modifying treatments for HD individuals. BDNF is a neurotrophin that is critically intertwined with an array of vital neuronal signaling pathways through TrkB. BDNF-TrkB signaling is strongly supportive of neuronal resilience, longevity, and normal function. The loss of BDNF in HD therefore provides both an explanation for the array of molecular, functional, and cognitive outcomes that manifest in HD as well as a potential pathway for treatment. Here, we extensively discussed multiple major HD mechanisms that have a shared outcome of neural BDNF reduction. We provided a special focus on the epigenetic and transcriptional dysregulation of BDNF in HD, a critical characteristic of its early molecular pathogenesis. Finally, we offered a critical analysis of emerging therapeutic approaches involving this neurotrophin within HD. Despite the promise of neurotrophin-based therapeutic approaches, there unfortunately remain major practical obstacles for their future use. These include delivery to deep neuroanatomical structures, poor pharmacokinetic profile, and uncertain efficacy in humans. Emerging viral-based strategies, Trk agonists, and brain-penetrant small molecules or peptides may soon overcome these challenges. Ultimately, we propose that neurotrophin therapies will likely form part of a multi-pronged series of interventions for HD involving lifestyle modification, symptomatic treatment, and small molecule-based approaches to modify disease progression. 

## 8. Search Methods and Strategy

Sources included in this literature review were populated from three major literature databases: Pubmed, Web of Science (Thompson Routers), and Scopus. These databases were supplemented from multiple gray literature sources, including Google Scholar, clinicaltrials.gov, EU Clinical Trials Register, and the bioRxiv preprint server. Our search strategy involved Boolean strings constructed from MeSH terms (e.g., “Brain-Derived Neurotrophic Factor” [Mesh] AND “Huntingtin Protein” [Mesh] AND “Axonal Transport” [Mesh]). These search strings were complemented by plain text searching within the MeSH string results, the examination of cross-referenced sources, and personal knowledge of relevant articles. Each of the authors independently executed these literature searches and prioritized articles that were published in the previous 5 years. There were no a priori inclusion or exclusion criteria.

## Figures and Tables

**Figure 1 biomedicines-11-02275-f001:**
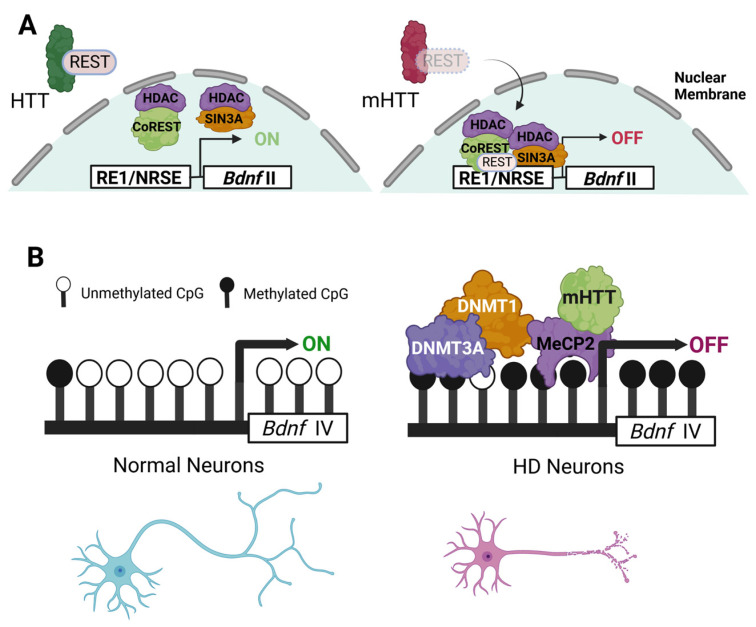
Selected mechanisms for *BDNF* gene dysregulation in HD. *BDNF* gene regulation is altered in HD and contributes to neuronal death through several mechanisms: (**A**) Wild-type HTT protein binds with REST in the cytoplasm, which hinders REST nuclear translocation and results in increased neuronal survival, synaptic plasticity, and neurite growth [155]. The reduced affinity of REST to mHTT allows the formation of a repressor complex on cis-regulatory elements of the *BDNF* exon II transcription start site. **CoREST**, REST corepressor 1; **HDAC**, Histone deacetylase; **NRSE**, Neuron-restrictive silencing element; **RE1**, Repressor element 1; **REST**, RE1-silencing transcription factor; **SIN3A**, SIN3 transcription regulator family member A. (**B**) The epigenetic dysregulation of *BDNF* gene transcription in HD neurons due to differential methylation patterns between normal and HD neurons. DNA methyltransferases (DNMT3A and DNMT1) methylate the regulatory elements of the *Bdnf* exon IV, resulting in the downregulation of its transcription in mHTT-expressing cortical neurons [104,105]. In HD neurons, gene repressive complexes (not shown) may be recruited to methylated cytosine through the action of mHTT and methylcytosine binding protein 2 (**MeCP2**) to silence *Bdnf* gene transcription [160].

**Figure 2 biomedicines-11-02275-f002:**
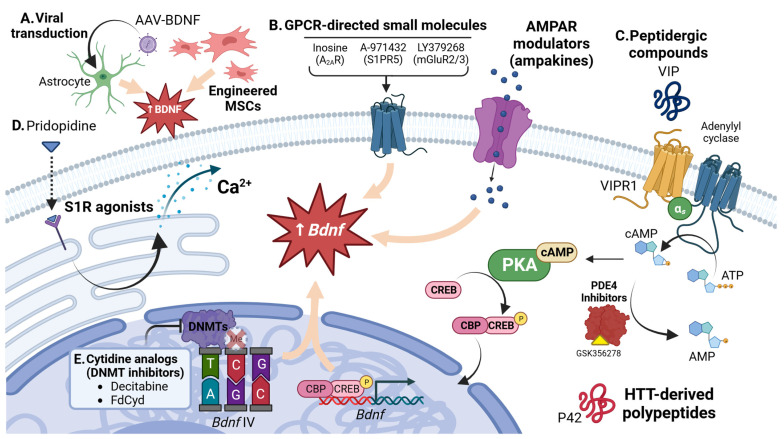
Overview of selected BDNF-directed molecular therapies in HD. Current therapeutic strategies for increasing BDNF expression in HD include (**A**) AAV-BDNF transduction of astrocytes or engraftment of BDNF-expressing MSCs; (**B**) small molecules targeting cell-surface G protein-coupled receptors (GPCRs) or ionotropic glutamate receptors; or (**C**) novel peptidergic compounds or cell-permeable small molecules that increase PKA signaling and CREB phosphorylation. Other approaches include (**D**) the modulation of ER calcium release or (**E**) the inhibition of DNMTs. Abbreviations used in alphabetical order: **A_2A_R**, Adenosine-2A receptor; **AAV**, Adeno-associated virus; **AMP**, Adenosine monophosphate; **AMPAR**, α-amino-3-hydroxy-5-methyl-4-isoxazolepropionic acid receptor; **ATP**, Adenosine triphosphate; **α_s_**, G-protein stimulatory alpha subunit; **cAMP**, Cyclic adenosine monophosphate; **CREB**, cAMP response element-binding protein; **CBP**, CREB-binding protein; **DNMT**, DNA cytosine-5-methyltransferases; **MSCs**, Mesenchymal stem cells; **Me**, Methyl group; **mGlu2/3**, Metabotropic glutamate receptor 2/3; **PDE4**, Phosphodiesterase 4b; **PKA**, protein kinase A; **S1PR5**, Sphingosine-1-phosphate receptor 5; **VIP**, Vasoactive intestinal peptide; **VIPR1**, Vasoactive intestinal peptide receptor 1.

## Data Availability

No new data were created or analyzed in this study. Data sharing is not applicable to this article.

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
