# Peer review of "Brain-Derived Neurotrophic Factor Dysregulation as an Essential Pathological Feature in Huntington’s Disease: Mechanisms and Potential Therapeutics"

_biomedicines, 2023, doi:10.3390/biomedicines11082275_

Round 1
Reviewer 1 Report
4 July 2023
Manuscript ID: biomedicines-2509715
Type: Review
Title: “Brain-derived neurotrophic factor in Huntington's disease: Recent advances and future directions” by Speidell A et al., submitted to Biomedicines
Dear Authors,
One of the current challenges is to create brain-derived neurotrophic factor (BDNF)-directed therapies for Huntington's disease (HD) that are efficient and can circumvent the shortcomings of available therapies. In the present review article entitled "Brain-derived neurotrophic factor in Huntington's disease: Recent Advances and Future Directions," Speidell and colleagues discuss therapeutic approaches focused on BDNF for the treatment of HD, a neurodegenerative disorder. Here, the authors explore various BDNF-focused therapeutic strategies, including BDNF supplementation and stem cell transplantation. The authors further discuss indirect approaches to up-regulate BDNF expression, such as exercise enrichment and environmental modification and enrichment (EME), highlighting how interventions have shown some benefits in preclinical models but may have limited therapeutic potential in advanced stages of HD. Additionally, the authors explore BDNF-targeting therapeutics, including repurposing small-molecule compounds and peptidergic compounds that modify BDNF expression or signaling. The review article concludes by emphasizing the challenges associated with delivering BDNF to deep neuroanatomical structures, pharmacokinetics, and efficacy in humans. But using viral-based delivery, small molecules that can get into the brain, BDNF agonists, and peptidergic compounds gives hope that these problems can be solved and that effective BDNF-based treatments for HD and other neurodegenerative diseases can be made.
The manuscript's main strength is that it addresses a timely and fascinating topic and presents a comprehensive and up-to-date review of the latest research on the role of BDNF in HD, including potential therapeutic strategies and future directions for the field.
In general, I think the idea of this article is really interesting, and the authors’ fascinating observations on this timely topic may be of interest to the readers of Biomedicines. However, some comments, as well as some crucial evidence that should be included to support the author’s argumentation, needed to be addressed to improve the quality of the manuscript, its adequacy, and its readability prior to its publication in the present form. My overall judgment is to publish this paper after the authors have carefully considered my suggestions below.
Please consider the following comments:
1. According to the Journals’ guidelines, I would suggest the authors use the Biomedicines Microsoft Word template file [1].
2. Title: This is the most important section of the manuscript. Please present a concise and self-explanatory title stating the most important message of this review. Suggestions: "Unleashing the Potential of Brain-Derived Neurotrophic Factor: Advancements and Prospects in Huntington's Disease Treatment"; " Brain-Derived Neurotrophic Factor: A Promising Therapeutic Avenue for Huntington's Disease and Beyond"; "Harnessing Brain-Derived Neurotrophic Factor for Neurodegenerative Disorders: Insights from Huntington's Disease" [2–4].
3. Abstract: I suggest the authors present the background, a succinct summary, and a conclusion in proportional order within the 200 words. The general background (one to two sentences), the specific background (two to three sentences), and the current issue covered by this review (one sentence) should all be included in the background before moving on to the objectives. I would like the author to provide background information, a problem statement, and their reasoning for branching off in this subsection. The brief review section concludes with a phrase that places this subsection in a broader context. The conclusion should begin with one sentence that summarizes the main message using words like "Here we highlight." The authors should describe the potential and the advancement this study has made in the field in the first sentence of the conclusion, followed by two to three sentences that provide a broader perspective that is easily understood by a scientist from any discipline [5–8].
4. Keywords: Please list ten keywords chosen from Medical Subject Headings (MeSH) and use as many as possible in the title and in the first two sentences of the abstract [9].
5. A graphical abstract that will visually summarize the main findings of the manuscript is highly recommended.
6. Introduction: I would like the authors to reorganize this section with about 1000 words and several paragraphs, introducing information on the key study constructs that should be understood by readers in any discipline, and make it persuasive enough to advance the primary goal of the author's recent research and the particular goal the author has intended by this review. I would like to suggest that the authors present the introduction beginning with the overall context, moving on to the specific context, and concluding with the current problem addressed in this review before moving on to the objectives. Those key structures ought to be set up logically and coherently [10].
7. In this regard, I believe that it would be helpful to provide a brief overview of the neural substrates involved in HD. HD is characterized by the progressive degeneration of specific neural substrates within the brain, primarily affecting the basal ganglia and cortical regions. These neural substrates play crucial roles in motor control, cognition, and emotion regulation (DOI: 10.3389/fpsyt.2023.1225755; DOI: 10.3390/ijms24044114). Additionally, the striatum, a key component of the basal ganglia, undergoes extensive neuronal loss in HD, particularly in the medium spiny neurons. This loss leads to disrupted signaling within the cortico-striatal pathways, resulting in motor impairments and cognitive dysfunction (https://doi.org/10.3389/fpsyt.2022.1054380). Furthermore, cortical regions, including the prefrontal cortex, motor cortex, and somatosensory cortex, also experience neuronal degeneration in HD (DOI: 10.17219/acem/165944). The involvement of these neural substrates highlights the multifaceted nature of HD and its impact on various brain regions crucial for normal functioning, and understanding the neural substrates affected by HD is essential for developing targeted therapeutic approaches aimed at mitigating the disease's symptoms and progression. In addition, the following works may enhance the value of this manuscript, including but not limited to: https://doi.org/10.3390/brainsci12050676; https://doi.org/10.3390/ijms21249338; https://doi.org/10.3390/ijms222212499; https://doi.org/10.3390/biomedicines11020235; doi: 10.17219/acem/154955; https://doi.org/10.3390/biomedicines10061368.
8. Neuronal Atrophy in HD: I suggest including a sentence or two explaining the normal function of the huntingtin protein and how the mutant form with the expanded repeat leads to neuronal atrophy. Also, I believe that the statement about the correlation between CAG repeat length and age of onset could be supported by referencing relevant studies or providing specific examples.
9. Movement Dysfunction and Regional Vulnerability in HD: This section highlights the selective loss of medium spiny neurons (MSNs) in the caudate and putamen, as well as the role of MSNs expressing the D2 dopamine receptor in generating the chorea characteristic of HD. Although the mention of other factors that may affect MSN vulnerability is informative, it would be helpful to provide a brief explanation or examples of these factors for better clarity.
10. Molecular Features and Impairments in HD: This section provides a good introduction to the molecular features associated with mutant huntingtin expression and their impact on neuronal health and function, but I would suggest to briefly explain the importance of each molecular feature mentioned (regulation of gene expression, translation, autophagy, mitochondrial function, intraneuronal transport, and synaptic function) and how their impairments contribute to HD pathogenesis.
11. BDNF and TrkB-BDNF Signaling: This section provides a clear introduction to BDNF and its role as a neurotrophin in neuronal development, survival, and function. Still, I recommend better explaining the specific functions of BDNF and TrkB-BDNF signaling, such as promoting neuronal growth and anti-apoptotic signaling, as well as their relevance to HD pathogenesis.
12. Discussion: I would like the authors to reorganize this section by opening with an introductory paragraph and followed by the summary of the previous section (Results). Then, I expect the authors to develop arguments clarifying the potential of this study as an extension of the previous work, the implication of the findings of this study, how this study could facilitate future research, the ultimate goal, the challenge, the knowledge and technology necessary to achieve this goal, the statement about this field in general, and finally the importance of this line of research. It is particularly important to present the limits, merit, and potential translation of this study to clinical practice [11,12]. The discussion of engineered mesenchymal stem cell (MSC) transplants is informative, but it would be useful to mention potential drawbacks or limitations of this approach, such as immune responses or ethical considerations.
13. References: The authors should consider revising the bibliography, as there are several incorrect citations. Indeed, according to the Journal’s guidelines, they should provide the abbreviated journal name in italics, the year of publication in bold, and the volume number in italics for all the references. Also, some of the references are out of date; please cite references from the last 10 years, particularly references from the last 5 years.
14. Conclusion: I think that presenting the conclusion would benefit from a single paragraph presenting some thoughtful as well as in-depth considerations by the authors as experts to convey the take-home message. The authors should make an effort to explain the theoretical implications as well as the translational application of their research. I believe that it would be necessary to discuss theoretical and methodological avenues in need of refinement as well as suggestions for a path forward in understanding the importance of this study.
15. I recommend that you submit your work to a native English speaker for assistance with the numerous grammatical errors that can be found in various sections of the manuscript.
Overall, the manuscript contains two figures, no tables, and 232 references. I believe that the manuscript may have important value in presenting a comprehensive and up-to-date review of the latest research on the role of BDNF in HD, its clear and concise presentation of complex scientific concepts, and its practical insights into the development of BDNF-based therapies for HD. Additionally, the manuscript provides a thorough analysis of the limitations of current treatment options and highlights the need for more effective BDNF-directed therapies. I hope that, after these careful revisions, the manuscript can meet the journal’s high standards for publication. I am available for a new round of revisions to this article. I hope that, after these careful revisions, this paper can meet the journal’s high standards for publication. I am available for a new round of revisions to this article.
I declare no conflict of interest regarding this manuscript.
Best regards,
Reviewer
References:
- https://www.mdpi.com/journal/biomedicines/instructions
- https://plos.org/resource/how-to-write-a-great-title/
- https://www.nature.com/nature-index/news-blog/how-to-write-a-good-research-science-academic-paper-title
- https://www.indeed.com/career-advice/career-development/catchy-title
- https://www.scribbr.com/dissertation/abstract/
- https://writing.wisc.edu/handbook/assignments/writing-an-abstract-for-your-research-paper/
- https://doi.org/10.5812/ijem.100159
- https://doi.org/10.4103/sja.SJA_685_18
- https://meshb.nlm.nih.gov/
- https://dept.writing.wisc.edu/wac/writing-an-introduction-for-a-scientific-paper/
- https://doi.org/10.3163/1536-5050.103.2.001
- https://www.scribbr.com/dissertation/discussion/
4 July 2023
Manuscript ID: biomedicines-2509715
Type: Review
Title: “Brain-derived neurotrophic factor in Huntington's disease: Recent advances and future directions” by Speidell A et al., submitted to Biomedicines
Dear Authors,
The manuscript exhibits generally clear language, although it contains a few grammatical errors and awkward phrasing that necessitate moderate editing. Enhancing sentence conciseness and organization, as well as incorporating lists and paragraphs, would significantly improve readability. While the paragraph displays a basic level of English proficiency, with some revisions, it can be enhanced to achieve a higher level of proficiency.
Best regards,
Reviewer
Reviewer 2 Report
In the manuscript “Brain-derived neurotrophic factor in Huntington's disease: Recent advances and future directions” the authors reviewed the diverse set of mechanisms triggering BDNF dysregulation within Huntington’s disease with a special focus on transcriptomic and epigenetic alterations. They highlight the importance and potential of BDNF as a promising therapeutic target for HD.
Overall, the review conveys a very good level of knowledge on the chosen topic.
I thank the authors for this review. I liked to read the review and I have significantly expanded my knowledge.
I would have liked to see the review in the respective template. That makes it easier to read the text including the figures.
Conclusion: It is somewhat surprising to me as a reviewer that I have no useful comments or suggestions for improvement on the manuscript. Except for the non-using of the template.
Reviewer 3 Report
It is very interesting to read the comprehensive review entitled “Brain-derived neurotrophic factor in Huntington's disease: Recent advances and future directions”. I really appreciate the authors effort for this wonderful piece of manuscript which might be considered for publication in the "Biomedicines” journal after the minor corrections mentioned below.
1. In the abstract, the authors just provided the background information, objective and overall summary of their findings. The method of data collection must be included after the objective (Line 40).
2. Further in the main text also, after the introduction, the author could emphasize the method of data collection which will be useful for the readers to get it know how and which data base the information has been obtained.
3. Lastly, in the conclusion, the authors needed to emphasize more on the significance of this review. This part should be made stronger using the insights and findings from the literature. Also provide future directions.
Round 2
Reviewer 1 Report
31 July 2023
Manuscript ID: biomedicines-2509715
Type: Review
Title: “Brain-derived neurotrophic factor in Huntington's disease: Recent advances and future directions” by Speidell A et al., submitted to Biomedicines
Dear Authors,
I am pleased to see that the authors have attempted to revise the manuscript. Nevertheless, the revisions remain partial in regard to my previous report. Prior to publication, I respectfully request that the authors consider my comments and revise the manuscript to meet the high standards of the journal. In addition, I anticipate the authors preparing a rebuttal to my remarks.
1. Title: Please avoid using abbreviation in the title.
2. Discussion: I would like the authors to present the discussion section as an independent section. I expect the authors to develop arguments clarifying the ultimate goal, the challenge, the knowledge and technology necessary to achieve this goal, the statement about this field in general, the importance of this li, and finally the significance of this li after an introductory paragraph and a summary of the previous sections. It is particularly important to present the limits, merit, and potential translation of this review to clinical practice [1,2].
3. References: Please follow the journal’s guidelines [3]. Journal abbreviations are punctuated with periods; page numbers end with periods; and no periods after doi numbers.
Overall, the manuscript contains two figures, no tables, and 247 references. I believe that the manuscript may have important value in presenting a comprehensive and up-to-date review of the latest research on the role of brain-derived neurotrophic factor in Huntington's disease. I hope that, after these careful revisions, the manuscript can meet the journal’s high standards for publication. I am available for a new round of revisions to this article. I hope that, after these careful revisions, this paper can meet the journal’s high standards for publication. I am available for a new round of revisions to this article.
I declare no conflict of interest regarding this manuscript.
Best regards,
Reviewer
References:
- https://doi.org/10.3163/1536-5050.103.2.001
- https://www.scribbr.com/dissertation/discussion/
- https://www.mdpi.com/journal/biomedicines/instructions
31 July 2023
Manuscript ID: biomedicines-2509715
Type: Review
Title: “Brain-derived neurotrophic factor in Huntington's disease: Recent advances and future directions” by Speidell A et al., submitted to Biomedicines
Dear Authors,
Based on the English proficiency assessment, it is noted that minor editing of the English language is required. While the overall communication is clear and understandable, there are some areas that could benefit from slight improvements in grammar, syntax, and word choice. Attention to detail, such as refining sentence structure and ensuring proper tense usage, will enhance the overall coherence and fluency of the written work. With some minor editing adjustments, the English language proficiency can be further enhanced.
Best regards,
Reviewer
Round 3
Reviewer 1 Report
8 August 2023
Manuscript ID: biomedicines-2509715
Type: Review
Title: “Brain-derived neurotrophic factor in Huntington's disease: Recent advances and future directions” by Speidell A et al., submitted to Biomedicines
Dear Authors,
I am pleased to see that the authors have addressed the issues I raised in the previous round. Currently, the manuscript is a well-written research paper with informative layouts that present a comprehensive and up-to-date review of the latest research on the role of brain-derived neurotrophic factor in Huntington's disease. I believe the manuscript meets the journal’s high standards for publication. I am looking forward to seeing more papers written by the same authors.
Thank you!
I declare no conflict of interest regarding this manuscript.
Best regards,
Reviewer
8 August 2023
Manuscript ID: biomedicines-2509715
Type: Review
Title: “Brain-derived neurotrophic factor in Huntington's disease: Recent advances and future directions” by Speidell A et al., submitted to Biomedicines
Dear Authors,
After evaluating the English proficiency, it has been determined that some minor revisions to the English language are necessary. While the overall communication is clear and understandable, certain areas could benefit from slight improvements in grammar, syntax, and word choice. Paying attention to detail, such as refining sentence structure and ensuring proper tense usage, will enhance the coherence and fluency of the written work as a whole. Making minor editing adjustments can lead to an improvement in English language proficiency.
Best regards,
Reviewer